# Food Sources and Dietary Quality in Small Island Developing States: Development of Methods and Policy Relevant Novel Survey Data from the Pacific and Caribbean

**DOI:** 10.3390/nu12113350

**Published:** 2020-10-30

**Authors:** Emily Haynes, Divya Bhagtani, Viliamu Iese, Catherine R. Brown, Jioje Fesaitu, Ian Hambleton, Neela Badrie, Florian Kroll, Cornelia Guell, Anna Brugulat-Panes, Arlette Saint Ville, Sara E. Benjamin-Neelon, Louise Foley, Thelma Alafia Samuels, Morgan Wairiu, Nita G. Forouhi, Nigel Unwin

**Affiliations:** 1European Centre for Environment and Human Health, University of Exeter Medical School, Truro TR1 3HD, UK; E.C.Haynes@exeter.ac.uk (E.H.); C.Guell@exeter.ac.uk (C.G.); 2MRC Epidemiology Unit, University of Cambridge School of Clinical Medicine, Cambridge CB2 0QQ, UK; divya.bhagtani@mrc-epid.cam.ac.uk (D.B.); anna.brugulat@mrc-epid.cam.ac.uk (A.B.-P.); louise.foley@mrc-epid.cam.ac.uk (L.F.); Nita.Forouhi@mrc-epid.cam.ac.uk (N.G.F.); 3Pacific Centre for Environment and Sustainable Development, University of the South Pacific, Suva, Fiji; viliamu.iese@usp.ac.fj (V.I.); morgan.wairiu@usp.ac.fj (M.W.); 4George Alleyne Chronic Disease Research Centre, University of the West Indies, St. Michael BB11000, Barbados; catherine.brown@cavehill.uwi.edu (C.R.B.); ian.hambleton@cavehill.uwi.edu (I.H.); alafiasam@gmail.com (T.A.S.); 5Pacific Community (SPC), Suva, Fiji; jioje2017@gmail.com; 6Faculty of Food and Agriculture, University of the West Indies, St Augustine, Trinidad and Tobago; neela.badrie@sta.uwi.edu (N.B.); arlette.saintville@sta.uwi.edu (A.S.V.); 7Land and Agrarian Studies, DSI-NRF Centre of Excellence in Food Security, Institute for Poverty, University of the Western Cape, Cape Town 7535, South Africa; florian@plaas.org.za; 8Department of Health, Behavior and Society, Johns Hopkins Bloomberg School of Public Health, Baltimore, MD 21205, USA; sara.neelon@jhu.edu

**Keywords:** diet, nutrition, non-communicable diseases, food sources, farming, backyard gardening, small islands developing states

## Abstract

Small Island Developing States (SIDS) have high and increasing rates of diet-related diseases. This situation is associated with a loss of food sovereignty and an increasing reliance on nutritionally poor food imports. A policy goal, therefore, is to improve local diets through improved local production of nutritious foods. Our aim in this study was to develop methods and collect preliminary data on the relationships between where people source their food, their socio-demographic characteristics and dietary quality in Fiji and Saint Vincent and the Grenadines (SVG) in order to inform further work towards this policy goal. We developed a toolkit of methods to collect individual-level data, including measures of dietary intake, food sources, socio-demographic and health indicators. Individuals aged ≥15 years were eligible to participate. From purposively sampled urban and rural areas, we recruited 186 individuals from 95 households in Fiji, and 147 individuals from 86 households in SVG. Descriptive statistics and multiple linear regression were used to investigate associations. The mean dietary diversity score, out of 10, was 3.7 (SD1.4) in Fiji and 3.8 (SD1.5) in SVG. In both settings, purchasing was the most common way of sourcing food. However, 68% (Fiji) and 45% (SVG) of participants regularly (>weekly) consumed their own produce, and 5% (Fiji) and 33% (SVG) regularly consumed borrowed/exchanged/bartered food. In regression models, independent positive associations with dietary diversity (DD) were: borrowing/exchanging/bartering food (β = 0.73 (0.21, 1.25)); age (0.01 (0.00, 0.03)); and greater than primary education (0.44 (0.06, 0.82)). DD was negatively associated with small shop purchasing (−0.52 (95% CIs −0.91, −0.12)) and rural residence (−0.46 (−0.92, 0.00)). The findings highlight associations between dietary diversity and food sources and indicate avenues for further research to inform policy actions aimed at improving local food production and diet.

## 1. Introduction

Globally, over 69 million people live in Small Island Developing States (SIDS), a group of 58 countries and territories (Figure 1). Thirty eight SIDS are independent states and full members of the United Nations, and over three-quarters of these are low- or middle-income countries [1]. Twenty nine of the independent SIDS are situated in the Pacific (*n* = 13) and Caribbean (*n* = 16), and in addition to common social, economic and environmental vulnerabilities, they share challenges related to inadequate and unhealthy diets [2]. Available data suggest that all twenty-nine of them have a double burden of malnutrition, with a high prevalence (>35%) of overweight or obesity in adults and a high prevalence (>20%) of anaemia in women of reproductive age [2]. Several SIDS, including Haiti, the Solomon Islands and Vanuatu, also have a high prevalence (>20%) of childhood stunting [2].

The high burden of overweight and obesity in the Pacific and Caribbean is associated with some of the highest rates of type 2 diabetes found globally, from a prevalence in adults of 9% in Haiti to over 25% in Nauru [3]. The high prevalence of diabetes is indicative of a high probability of premature mortality from non-communicable diseases (NCDs), and more than 1 in 5 adults are expected to die from an NCD in most SIDS before their 70th birthday [4].

One of the factors associated with these high burdens of diet-related conditions is an increasing reliance on food imports, and a decline in the consumption of locally produced foods [5]. For example, the contribution of imports to consumed food rose between 1990 and 2011 from less than 45% to 60% in the Pacific and to over 67% in the Caribbean [6]. This increasing reliance on food imports was partly driven by trade liberalisation policies of the 1990s, resulting in widespread decline in export-oriented agricultural sectors, and contributing to general declines in national food production [7]. The imported foods tend to be energy dense, highly and ultra-processed and of low nutritional quality [5].

In 2014, SIDS governments agreed to the goal of ending malnutrition in all its forms by 2030 [8]. The Global Action Programme on Food Security and Nutrition in Small Island Developing States [5] provides a framework around three broad areas: building enabling environments, developing sustainable and resilient food systems, and empowering people and communities for food security and nutrition. The Global Action Programme promotes coordinated and complementary interventions across entire food systems, rather than sector specific interventions [5]. This system-wide approach includes promoting interventions to increase local food production that are socially and economically empowering, and environmentally sustainable. The Global Action Programme addresses the economic, social, and nutritional needs of vulnerable sections of the population, including smallholder and home-based food producers. An important concept related to its aim is that of ‘food sovereignty’, and that is the ability of populations to shape their own food systems towards culturally appropriate foods that are sustainably produced and health promoting [9].

Recent systematic and scoping reviews [10,11,12,13,14] demonstrate the need for better-quality evidence on the links between food production interventions in low- and middle-income countries (LMICs) and a range of potential impacts in order to inform policy. This includes the need for more evidence on the links between where people source their food, the quality of their diets and health. We have shown as part of the work for this study described here, the Community Food and Health (CFaH) project, that there is a particular lack of evidence on these links from SIDS [12]. Our ultimate goal in CFaH is to contribute to filling these evidence gaps in support of policy initiatives, and towards this we aimed to develop methods for evaluating the impacts of community-based food production initiatives and to use them to collect data to inform further work.

Our specific objectives were to (i) develop a toolkit of methods for assessing the relationships between diet, sources of food consumed, and the risk of nutrition-related NCD; and (ii) to investigate associations between individual socio-demographic characteristics, food sources, and aspects of diet, including dietary diversity in two SIDS: Fiji and St Vincent and the Grenadines (SVG). Additionally, we considered the strengths and limitations of our methods and how to improve them for application in the evaluation of interventions aimed at increasing local food production and nutrition.

## 2. Materials and Methods

### 2.1. The Settings and Study Populations

The funding for this project required that it be undertaken in countries eligible for official development assistance [16]. We conducted this study in two upper middle-income SIDS [16], one located in the Pacific (Fiji), and another in the Caribbean (SVG) (see Figure 1 for their location). The choice of these two countries was largely pragmatic. The research team in the Pacific is based in Fiji, and the research team in the Caribbean, based in Barbados (a high-income country), has good links with the Ministry of Health in relatively nearby SVG. Fiji is one of six upper middle-income independent SIDS in the Pacific, the other independent SIDS in the Pacific being lower middle (*n* = 2) or low income (*n* = 5). SVG is one of eleven upper middle-income independent SIDS in the Caribbean, the other independent SIDS in the Caribbean being high income (*n* = 4) or low income (*n* = 1) [16].

In 2010, Fiji had a population of 875,000 and a land area of 18,000 km^2^. SVG is smaller, with a population of approximately 109,000 and a land area of 400 km^2^. Both countries have high and increasing burdens of diet-related NCDs. Recent estimates of obesity prevalence in men and women in Fiji are 25% and 35%. In SVG, the estimated prevalence of obesity is 17% and 31% in men and women, respectively [17]. Estimates of diet-attributable mortality and morbidity in these two countries are high. For example, Fiji has an estimated 636 diet-attributable deaths per 100,000 per year, and SVG has an estimated 264 diet-attributable deaths per 100,000 per year, compared to 121 per 100,000 per year in Western Europe [18].

Within each country, we purposively sampled to include urban and rural areas (based on enumeration districts) and cover a broad range of socio-economic conditions (urban, rural, high and low socio-economic status). We selected the areas in consultation with relevant national stakeholders, including the National Food and Nutrition Centre and Ministry of Rural and Maritime Development in Fiji; the government statistics office, responsible for the national census, in SVG; and the Ministries of Health in both countries. Individual households within these designated areas were selected to participate. In SVG, a convenience sample of all households within the selected areas were approached. In Fiji, households were numbered using satellite photographs and selected via a computer random number generator. In both settings, all household residents aged ≥15 years were eligible to participate in the survey. This age is widely used as the lower value for people of working age in demographic and health surveys and as the lower value for women of reproductive age (an important subgroup in our analysis plan) [19]. Appendix A indicates the approximate location of the study areas.

### 2.2. Survey Toolkit—Rationale, Sources and Development

The underlying rationale for the development of the toolkit was to draw together existing data collection instruments to provide a single resource for efficient ‘population-scale’ data collection in resource-limited settings that could be easily tailored to local environments. The research partners in the Pacific, Caribbean and the United Kingdom co-designed the toolkit, ensuring that it was tailored to suit the two settings, informed by evidence reviews [12] and agreed upon at an investigators workshop.

The toolkit comprised six questionnaires arranged in the following order to reduce the risk of biasing responses to the subjective participant-based dietary recall [20]: (1) initial questions on respondent’s age, sex, and the number and characteristics of other household members; (2) 24 h dietary recall and dietary diversity indicator; (3) dietary screener; (4) food sources questionnaire; (5) food insecurity experience scale questionnaire; and (6) additional demographic details, aspects of medical history and the recording of measured height, weight and blood pressure. We describe key sources and features of the data collection toolkit below.

### 2.3. Dietary Diversity

Dietary diversity is a qualitative measure of the range of foods consumed. The dietary diversity indicator tool, collaboratively designed by the Food and Agriculture Organization (FAO) and USAID’s Food and Nutrition Technical Assistance III (FANTA) project, comprises a data collection instrument and associated metric to indicate the range of foods consumed at household or individual level [19]. The data collection instrument can be tailored to individual settings but maintains the ability to create internationally comparable dietary diversity scores, defined as the number of standard food groups consumed over a 24 h reference period. This provides overall information on food consumption patterns and offers a quantitative indicator to draw comparisons within and between settings [19]. The dietary diversity score can also be used to indicate micronutrient adequacy for certain populations. There is no dichotomous indicator for minimum dietary diversity for all individuals. However, a standard cut off of ≥5 of the 10 defined food groups has been validated to reflect ‘minimum dietary diversity’ (as a proxy to micronutrient adequacy) for women of reproductive age (15 to 49 years) [21].

In line with the FANTA/FAO dietary diversity methods framework [19], we employed a multiple-pass open recall to capture the range of foods and drinks consumed over the 24 h of the previous day. We classified the foods into the recommended ten core food groups required to calculate the dietary diversity score and indicate minimum dietary diversity for women (of reproductive age): (1) grains, white roots and tubers and plantains; (2) pulses; (3) nuts and seeds; (4) dairy; (5) meat poultry and fish; (6) eggs; (7) dark green leafy vegetables; (8) other vitamin A-rich fruit and vegetables; (9) other vegetables; (10) other fruit. We also included eight non-essential groups, that were not part of the dietary diversity score calculation, to capture the full range of foods consumed given our wider application of the tool beyond women of reproductive age [19]: (1) insects and other small protein foods; (2) red palm oil; (3) other oils and fats; (4) savoury and fried snacks; (5) sweets; (6) sugar-sweetened beverages (SSB); (7) condiments and seasonings; (8) other beverages and foods. In addition, we separated the core ‘meat, poultry and fish’ group to enable separate descriptions of consumption of these three food types (and further delineated by level of processing), but converged these subgroups for the purpose of calculating the dietary diversity score. To contextualise our data collection tool, regional investigators contributed to the development of a food classification guide for local foods and common dishes and indicated how mixed dishes should be classified into food groups. Additionally, in line with the FANTA/FAO guidelines (19), we did not assess the portion size or quantity of each food consumed except to identify negligible amounts that are less likely to contribute to micronutrient adequacy, and therefore recorded whether the amount consumed was usually less than, or greater or equal to 15 g. Only items of which 15 g or more were consumed were sufficient to contribute to the dietary diversity score [19].

### 2.4. Diet Screener

A diet screener comprised a series of diet-related questions that aimed to capture the frequency and quantity of consumption of particular foods and beverages that are known to be associated with NCDs. Questions were informed by the diet component of the WHO STEPS survey [22] and regional and national dietary guidelines (Pacific/Fiji; Caribbean/SVG) in order to capture consumption patterns of salt, fats, fruit and vegetables, red meat, fish and sugar-sweetened beverages, and of foods according to their level of processing (e.g., unprocessed to ultra-processed items) [23]. The questions asked participants to recall how often they consume these types of foods (number of times in a typical week) and how much they consume (number of servings consumed on one of those days). We used measuring instruments and photographs of foods to encourage consistency in reporting by providing reference examples of food items and serving sizes for respondents and data collectors.

### 2.5. Food Sources

We designed questions on food source to capture how and where consumed foods were sourced and the extent to which those sources were utilised. The questions were centered on whether individuals source their food through their own production (growing, gathering, hunting or fishing), purchasing (from various retailers or food service business, as indicated in Table 1), borrowing/exchanging/bartering (BEB) or food aid; categories as recommended by the diet diversity framework [24] and research conducted in similar settings [25]. We obtained further details by asking standardised follow-up questions according to individual responses, aimed at determining the type of food sourced in given ways, and how frequently those foods were consumed. The questionnaire was designed so that food source data could be collapsed into the same food groups used in the diet diversity and diet screener questionnaires.

### 2.6. Health, Social and Demographic Characteristics

Questions on participant-reported age, sex, ethnicity, household size, education, and aspects of medical history (including treatment for hypertension) were based on WHO and PAHO STEPS survey instruments [22,26]. Height and weight were measured using a mobile stadiometer (Seca 2017 Hamburg, Germany) and digital scales (Seca Robusta 813) to calculate body mass index. Blood pressure was measured following protocols described in detail elsewhere [27] using a digital, automatic blood pressure monitor (Omron Tokyo, Japan, HEM-705CP/Mediscope, Basingstoke, UK). We measured blood pressure three times and used the means of the second and third systolic and diastolic measurements in analysis.

### 2.7. Data Collection

Data collection was conducted from August 2018 to November 2018. We trained two teams of data collectors from Fiji and SVG to administer all components of the questionnaire, including the open dietary recall for the previous day. In Fiji, all data collectors were local and fluent in iTaukei (Bauan dialetic) and English. In SVG, data collectors were also local. Data collectors used android tablets to enter data into an electronic version of the toolkit constructed in REDCap© (version 7.3.4, USA) [28]. The dietary recall data were initially recorded by hand onto a paper form. The data collector and respondent collaboratively reviewed the recalled items and the data collector classified each recalled item into a food group list in REDCap. Data collectors had access to bespoke resources, including photographic examples of local foods, and were trained to help respondents think through the individual food groups contributing to a dish. In Fiji, these resources were available in iTaukei and English, and interviews were conducted in iTaukei or English, as appropriate to the interviewee. In SVG, all interviews were conducted in English (which is the local language).

Data collectors in each country team benefited from their understanding of local cultural practices and in both settings some had previous experience in nutrition research or practice. Training materials were developed by project staff and made available online, including presentation slides, ‘how to’ videos, standard operating procedures and show-card examples of food items for each of the in-country teams. All data collectors undertook online anthropometric training, which included technical videos produced by the MRC Epidemiology Unit at the University of Cambridge. Additionally, data collectors and project staff in Fiji and SVG engaged in a series of virtual and actual face-to-face sessions to allow an opportunity for any uncertainties to be resolved, including in Fiji on correct iTaukei terms for questionnaire items, and to promote competence and consistency in anthropometric measurement. Finally, the survey was piloted within the research team, and on small numbers (10 to 20) in Fiji and in SVG, to check for ease of administration, relevance and appropriateness of the content to the setting, reliability and face validity, prior to data collection.

### 2.8. Ethics

The University of the South Pacific and University of West Indies ethics boards provided ethical approval in March and June 2018, respectively. The University of Cambridge Psychology Research Ethics Committee provided oversight by reviewing and endorsing the ethical approval given by the University of the South Pacific (in March 2018) and the University of the West Indies (in July 2018). In addition, the Ministries of Health of Fiji and SVG granted permission for the research to be conducted. All respondent provided written informed consent, with adolescent assent and accompanying guardian consent for respondents aged 15–18 years.

### 2.9. Sample Size and Statistical Analysis

#### 2.9.1. Sample Size

As described, our aim in this exploratory project was to develop contextually appropriate methods, collect preliminary data and undertake analysis that would help to inform future work. We aimed for moderate statistical precision on nutritional adequacy and, based on this pragmatic consideration, we planned to collect data from 100 households each in Fiji and SVG (i.e., 200 households in total). We estimated that this would provide a 95% confidence interval of +/−8% on a proportion of 50% assuming a design effect of 1.5 [29].

#### 2.9.2. Statistical Analysis

We performed all analyses using Stata statistical software (version 13, Stata Corp, College Station, TX, USA). We present characteristics of the study populations as proportions, medians (IQR) or means (SD). We examined bivariate associations between regular (>weekly) versus less regular (≤weekly) use of a food source and socio-demographic characteristics as the difference (in mean, median or proportion) with 95% confidence intervals. The confidence intervals on the difference between two medians were estimated using quantile regression. We examined associations between food source and aspects of diet in the same way. Generally, we treated the dietary diversity score as a continuous variable, with a range of 0 to 10; however, for women of reproductive age (15 to 49 years), we describe the proportion meeting minimum dietary diversity [19].

We used ordinary least squares multiple linear regression to identify statistically independent associations between the dietary diversity score (dependent variable), food sources and socio-demographic factors. In this analysis, we combined Fiji and SVG data, with an independent variable for country. We only included food sources that were associated with the dietary diversity score (i.e., where the 95% CIs on the difference did not cross zero) in at least one country, and only included demographic variables that were associated with food sources using the same criterion. Finally, we examined the associations between demographic characteristics, food sources, aspects of diet and overweight or obesity (BMI ≥ 25 kg/m^2^), and hypertension. We defined hypertension as a measured blood pressure of ≥140 mmHg systolic and/or ≥90 mmHg diastolic; and/or taking medication for hypertension. We used logistic regression to identify independent associations with overweight or obesity and hypertension, respectively. Given the exploratory nature of this, study we did not adjust for household clustering or multiple testing in the bivariable analyses, but we did always adjust for household clustering in our multivariable regression analyses that inform our conclusions.

## 3. Results

### 3.1. Characteristics of the Study Populations

In Fiji, we recruited 186 individuals aged 15 years and over from 95 households; in SVG, we recruited 147 individuals from 86 households (Table 2). In both countries, almost two-thirds (63%) of respondents were female, and half (49%) were between 15 and 40 years of age. In Fiji, the large majority (78%) of respondents identified as iTaukei, and most others (21%) identified as Indo Fijians. In SVG, half (52%) of the respondents identified as Black or African origin, just under one-third (31%) as Amerindian (indigenous peoples of the Americas), and most of the rest (17%) as ‘mixed’. Approximately half of all respondents had completed primary education only, with 14% having completed tertiary education. The prevalence of overweight or obesity, defined as a BMI ≥25 kg/m^2^, was high; 70% in Fiji and 65% in SVG. Hypertension, defined as taking anti-hypertensive medication and/or raised blood pressure (≥140/90 mmHg) was 39% in Fiji and 28% in SVG.

### 3.2. Dietary Diversity and Aspects of Diet

Proportions reporting the consumption of different food groups in the 24 h dietary recall in Fiji and SVG are shown in Figure 2. Notable differences (all *p* values < 0.01) include the higher proportions consuming sugar-sweetened beverages (SSBs) and meat and poultry in SVG compared to Fiji, and higher proportions consuming fish/seafood and vegetables in Fiji compared to SVG. The mean dietary diversity score, which is based on the ten food categories in groups b, c, and d in Figure 2, was less than 4 in both rural (3.5) and urban (3.7) Fiji, and was 3.6 and 4.2 in rural and urban SVG (Table 3). The proportion of women of reproductive age meeting minimum dietary diversity (a proxy for micronutrient adequacy) ranged from 23% in urban Fiji to 43% in urban SVG. Median weekly servings of fruit and vegetables ranged from 15 in rural Fiji to 6 in urban SVG. Median weekly servings of SSBs were three in Fiji and seven in SVG.

### 3.3. Sources of Food and Associated Socio-Demographic Characteristics

#### 3.3.1. Purchasing

The vast majority of respondents (97% Fiji; 93% SVG) reported sourcing at least some of the food that they consume through purchasing (Appendix A). Purchased food from a supermarket or wholesaler was consumed more than once a week by 59% of participants in Fiji and 71% in SVG (Table 4 and Table 5). The only socio-demographic characteristic clearly associated with regular consumption of food from supermarkets was rural residence in Fiji (Table 4), with tendencies in both countries to those with larger household size and greater than primary education, although the 95% CIs on the difference included zero (Table 4 and Table 5).

Greater than weekly (regular) consumption of foods sourced from smaller shops was reported by 22% of respondents in Fiji and 53% in SVG. Consumption from formal small shops was reported by 12% in Fiji and 37% in SVG, and informal small shops by 19% and 31%, respectively. In SVG, characteristics associated with regular consumption from a small shop were: rural residence for formal shops, and female sex and greater household size for informal shops. Finally, regular consumption of food from food service businesses (e.g., food from takeaways, and restaurants) was relatively uncommon, being reported by approximately 7% of respondents in both countries, and was also associated with younger age, particularly in SVG (28 vs. 42 years) (Table 5).

#### 3.3.2. Own Production

A majority of respondents (83% Fiji; 56% SVG) reported sourcing some of their food from their own production (Appendix A), with 68% in Fiji and 45% in SVG consuming their own produce more than once a week (Table 4 and Table 5). This was associated with rural residence in both countries, most strongly in SVG, where 88% of those consuming their own produce more than weekly were rural residents. In Fiji, own production was associated with household size, where 69% of those consuming their own produce less than weekly lived in households with more than three people (Table 4).

#### 3.3.3. Borrowing/Exchanging/Bartering (BEB)

Just under one-quarter (23% Fiji) to over one-third of respondents (40% SVG) said that they sourced some food through BEB. However, only 5%, in Fiji reported greater than weekly consumption of BEB food, compared to 33% in SVG. In both countries, the types of food most commonly sourced in this way included staples (roots/tubers/plantains) and vegetables, as well as fish in Fiji (Appendix A). In Fiji, regular consumption of food sourced by BEB was associated with older age and smaller household size (Table 4). In SVG, there was no difference in age, but regular consumption of BEB food was associated with greater than primary education and rural residence, and marginally with female sex (Table 5).

#### 3.3.4. Food Aid

Having ever sourced food aid was reported by a small minority of respondents—5% in Fiji and 2% in SVG. In both Fiji and SVG, this was associated with older age and female sex, although the small numbers precluded further analysis (Appendix A).

### 3.4. Associations between Different Food Sources

Regular (greater than weekly) use of some food sources (as listed in Table 1) was strongly associated, and these associations differed somewhat between Fiji and SVG (Appendix A). In Fiji, but not in SVG, regular consumption of food from a formal and informal small shop was positively associated with consumption from a food service business. In SVG, but not in Fiji, the regular consumption of BEB food was positively associated with regular consumption from own production and supermarket purchases.

### 3.5. Associations between Food Sources and Aspects of Diet

There were associations between regular food consumption from different sources and aspects of dietary quality in both Fiji and SVG.

In both countries, there was an association between regular BEB food consumption and a higher dietary diversity score (Table 6 and Table 7). BEB sourcing was also associated with greater fruit and vegetable consumption in both countries. Regular consumption of own produce was also associated in both countries with greater fruit consumption, but not with other aspects of diet.

Regular consumption of food from a supermarket was associated with positive aspects of diet in both countries, with higher median reported intakes of fruit and vegetables, and with greater dietary diversity in SVG (Table 6 and Table 7). However, in SVG, it was also associated with higher reported weekly intakes of SSBs and red and processed meat (Table 7).

Regular consumption of food from a small shop was associated in both countries with substantially higher intakes of SSBs, and with higher intake of red or processed meat (but with 95% confidence intervals CIs crossing zero in SVG for the formal small shop). Regular consumption of food from a food service business was associated in both countries with SSBs and in Fiji with red and processed meat consumption. In Fiji, but not SVG, consumption from both a small shop and a food service business was associated with higher reported fruit consumption (Table 6), and consumption from a food service business was associated with higher reported vegetable consumption in SVG (Table 7).

### 3.6. Multivariable Analsyses

#### 3.6.1. The Dietary Diversity Score, Socio-Demographic Characteristics and Food Sources

In multiple linear regression, regular consumption of food from a small shop and rural residence were negatively associated with the dietary diversity score (Table 8). Conversely, regular BEB food sourcing, education greater than primary, and older age were positively associated with the dietary diversity score (although the 95% CIs for age include 0, *p* = 0.063). In analyses stratified by rural vs. urban residence, regular consumption of BEB food was positively associated in both areas with the dietary diversity score (data not shown).

#### 3.6.2. Overweight/Obesity, Hypertension and Socio-Demographic Characteristics, Food Sources and Aspects of Diet

Bivariate associations between overweight or obesity, hypertension and socio-demographic characteristics, food sources, and aspects of diet are presented in Appendix A. In both Fiji and SVG, older age and female sex were strongly associated with overweight or obesity, and age and less than secondary education with hypertension. In both countries, higher reported intakes of SSB and red or processed meat tended to be negatively associated with overweight or obesity in bivariate analyses. However, these relationships were not found in multivariable analysis (binary logistic regression, Appendix A) where only older age and female sex were associated with overweight or obesity and only older age with hypertension.

## 4. Discussion

The primary objective of the analyses in this study was to explore in two SIDS, Fiji and SVG, the relationships between where people source food, socio-demographic characteristics, and dietary quality. As far as we are aware, this is the first study from SIDS to explore these relationships concurrently [12], and one of a limited number to do so from other low- and middle-income settings [10,11,14]. Our findings are consistent with what is already known about diet and nutrition in middle-income SIDS of the Pacific and Caribbean. In addition, we found relationships that have not been previously described, further investigation of which could inform interventions aimed at improving diet and nutrition.

### 4.1. Aspects of Nutrition and Diet Compared to Previous Studies

The high prevalence of overweight or obesity, present in approximately two-thirds of our study populations and substantially higher in women than men, is in line with the previously published figures for Fiji and SVG from the NCD Risk Factor Collaboration and World Health Organization [4,17]. The prevalence of hypertension in our study, similar in women and men, is also consistent with other sources [4,17]. With regard to diet, low levels of fruit and vegetable intake and high intake of SSBs in both countries is also typical of similar SIDS in the Pacific and Caribbean [2]. Our data showed twice the intake of SSBs in SVG compared to Fiji, which is consistent with data from the Global Burden of Disease Study [2], and representative of differences between SIDS in the Pacific and Caribbean as a whole [2]. The Caribbean has some of the highest rates of consumption of SSBs globally [2].

A recent study from Fiji in rural indigenous food-producing households, from eight villages, assessed dietary diversity (DD) using the Household Dietary Diversity Score (which is recommended for use in agriculture-dependent areas). It does not include foods eaten outside the home, which our approach does, and is scored from 0 to 7, unlike ours, which is scored from 0 to 10. Given different methods and study populations, comparison is difficult, but one finding of theirs causes us to reflect on our approach. They describe a strong association between low DD and low farm diversity [30]. In our study, we did not assess the diversity of own food production, but it is certainly a hypothesis worth pursuing in future work that diversity of own production is positively related to diversity of consumption.

### 4.2. Borrowing, Exchanging or Bartering and Diet

A new and potentially important finding from our analyses is the independent association between regularly sourcing food through borrowing, exchanging, or bartering (BEB) and higher dietary diversity. Although BEB was much less common in Fiji, where only 5% of respondents reported regularly sourcing food in this way, compared to one-third of respondents in SVG, it was associated with higher dietary diversity in both countries. As far as we are aware this association between regular BEB of food and higher dietary diversity has not been previously reported. However, other aspects of food exchanging and sharing have been described in a variety of settings. In indigenous societies in Greenland and the Canadian Arctic, food-sharing networks have been described as providing access to traditional foods and as an important food source in times of stress [31,32,33]. In a study from a small community in North West Namibia, complex networks of reciprocal food exchange have been described that are based on both geographical proximity and interpersonal dynamics of trust [34]. Studies have also been undertaken in less traditional societies, such as with teenagers in Soweto, South Africa, describing how they pool and share resources to access what are typically less healthy foods [35]. There is also growing interest in the role of food BEB in urban settings, including in high-income countries. The Sharecity project [36], for example, recently documented the types of food BEB practices and arrangements in 100 cities globally [37]. In a separate study, it was estimated that use of one food-sharing app in London, UK, led to a substantial reduction in food waste [38].

These examples illustrate the heterogeneity of practices and arrangements that fall under the broad heading of ‘food from BEB’ and illustrate growing recognition of the importance of social relations to food security. We acknowledge that more work is required to gain greater understanding of the significance of our findings in Fiji and SVG. There are key questions, unanswerable from our current data, around the exact nature of BEB practices and the circumstances in which they are taking place, the types and extent of relationships between those involved, how such interactions have and are changing overtime, and whether there is potential for leveraging these approaches as part of improving dietary quality across the population. In addition, we cannot accurately determine from our data the original sources (e.g., whether purchased or own produced) of the food in BEB. However, our data do show that the foods most commonly sourced in this way were local staples of roots, tubers and plantains, followed by vegetables and fruit. Additionally, in SVG, but not in Fiji, regular BEB of food was strongly associated with consumption of own produced food. This suggests that in SVG a high proportion of BEB food was ‘own produced’.

While the BEB category undoubtedly captures different types of social interactions (e.g., borrowing by households in need, exchanging different foods across households with excess production, and bartering food for services/products by households), the findings suggest that households are employing their social assets to increase access to foods through non-monetary means. Understanding how these community-level responses function in lower-income communities may identify potential spaces for policy action around food insecurity. It is perhaps relevant to note that the latest Brazilian dietary guidelines emphasize the social importance of food, recommending that families share in the ‘acquisition, preparation, cooking’ and consumption of meals [39].

In Fiji, BEB has taken on a new importance in response to the economic disruption caused by the Covid-19 pandemic. For example, it is reported that over one in six Fijians have joined a Facebook group called ‘Barter for Better Fiji’ [40], which was established in response to the pandemic. A wide variety of goods are being traded, including home-grown and reared food items. The development of these organically formed groups highlights emerging collective action initiatives, guided by evolving rules, as home growers seek to source locally produced foods and connect with others outside of their communities. Further research should inform policy initiatives to promote the development and maintenance of such social food networks and promote resilience in the face of future shocks.

### 4.3. Own Food Production and Diet

Own production of food, such as backyard gardening, has been widely promoted as an effective approach to improving dietary quality. However, we have previously shown that from SIDS there is little published research evidence to support this [12], and evidence from other parts of the world is inconsistent [10,11,14]. In this current study, we found evidence of an association between regular consumption of own produced food and greater fruit consumption (in both Fiji and SVG), but no association between own produced food and dietary diversity. Clearly, a lack of association in our cross-sectional data does not rule out positive dietary benefits of home food production, beyond that of fruit consumption, but a well-controlled longitudinal study would be required to evaluate this.

### 4.4. Types of Retail Outlets and Diet

Although regular consumption of food from a small shop was independently associated with lower dietary diversity in our multiple linear regression analysis, it is relevant to note that in bivariate analyses regular consumption of food from a small shop, and indeed from a supermarket, was associated with both positive (e.g., fruit and vegetables) and negative (e.g., SSBs, red or processed meat) aspects of diet. This observation indicates the need for further research to better understand and inform policy on what underlies the availability, presentation and pricing of items within these retail outlets, and the types of policy interventions that could promote ‘nutrition sensitive value chains’ [41] and the purchasing and consumption of healthier items. Such research should include investigation of the effectiveness of regulatory, economic and infrastructural measures designed to promote the supply and consumption of healthy foods from small shops, while limiting the provision of ultra-processed and obesogenic foods.

### 4.5. Associations with Overweight/Obesity and Hypertension

In multivariable analyses, we found no statistically independent associations between any aspects of diet and overweight/obesity or hypertension. We did find negative bivariate associations between processed or red meat consumption and overweight/obesity in Fiji and SSB consumption and overweight/obesity and hypertension in SVG. While these findings appear contradictory at first sight, they appear to be largely due to confounding by age. These dietary items are more frequently consumed by younger adults, and overweight/obesity and hypertension are commoner in older adults, hence the associations are not apparent when controlling for age. It is also possible that ‘reverse causation’ contributes to the bivariate associations, if adults with those health conditions reduce red/processed meat and SSB consumption. A longitudinal study design would be required to properly investigate these relationships.

### 4.6. Strengths and Limitations

It is important to acknowledge the limitations and strengths of this study. This was a relatively small exploratory cross-sectional study, with purposive sampling designed to capture a breadth of socio-economic conditions in rural and urban Fiji and SVG. As such, we cannot claim that our data are nationally representative, although it is re-assuring that our prevalence estimates on aspects of diet, obesity and hypertension agree with national estimates. The relatively small sample size also limits our ability to undertake subgroup analyses. In terms of dietary quality, we used the pragmatic approach of assessing dietary diversity, plus the intake of specific food items. As described in the methods section, there are precedents for using dietary diversity as a proxy for dietary quality. However, a limitation is that in adults it has only been validated as an indicator of micronutrient adequacy in women of reproductive age.

Our questionnaire on food sources worked well in enabling us to assess the associations between frequency of use of food sources and aspects of diet but it could be improved. For example, we could not quantify the actual contribution of individual food sources to diet. One approach to achieving this could be to assess diet through a fully quantified dietary recall or diet diary including portion size information, and for each item to inquire about its source(s). Clearly, this would require a much heavier participant burden, and the feasibility of such an approach would need to be assessed. In addition, our current approach to collecting data on food sources is unable to determine whether food procured by purchasing is locally produced or imported. In order to determine this, another layer of data collection would be required, i.e., to trace value chains upstream to determine where retail outlets source their food items. This type of research, especially if undertaken with actors from across the food system, would further inform the design of policy instruments aimed at increasing the availability and consumption of nutritious local foods.

Finally, we also acknowledge that seasonality may have influenced our findings. Data collection took place during the southern hemisphere winter and northern hemisphere summer. Although both Fiji and SVG are tropical countries and seasons are much less pronounced than in temperate zones, summer months tend to be wetter and a few degrees warmer, and some fruits and vegetables grow seasonally [42,43]. Dietary diversity scores, therefore, may differ at different times of year, and it is possible that relationships with dietary diversity may also vary by season. Seasonality may also have influenced responses to the questions as to where people source food, even though we asked respondents to think of a ‘typical week in the last year’.

A strength of our study is that we developed and used data collection tools that were specifically designed to explore the relationships between food sources, aspects of diet, nutrition-related outcomes and socio-demographic characteristics in these two resource constrained settings. We believe that this is the first time this has been performed in SIDS settings. Moreover, we were able to conduct this study in two middle-income SIDS in different regions, thus providing unique comparative data between one country in the Pacific and one in the Caribbean, illustrating similarities and differences between them. It is worth noting that the data collection tools developed for this study have subsequently been adapted to assist in the evaluation of two Pacific Island food production projects of the Technical Centre for Agricultural and Rural Cooperation ACP-EU (CTA) [44].

## 5. Conclusions

In conclusion, we have developed methods and undertaken a study in two SIDS, both known to have high burdens of diet-related NCDs, to explore the relationships between aspects of diet, food sources and socio-demographic factors. As expected, we found a high prevalence of overweight/obesity and of hypertension. A novel finding is the positive relationship between BEB food sourcing and dietary diversity. The methods and findings from this study are being used to plan further policy relevant research [5] aimed at developing and evaluating interventions to improve nutrition through the increased production and consumption of local foods. One example is a study in which we are working closely with two well-established non-governmental organizations (NGOs), one in Fiji and one in SVG. In this study, we will evaluate the nutritional, social and economic impacts of food production interventions delivered by the NGOs [45] and thus contribute towards strengthening the evidence base for improving food sovereignty and reducing burdens of malnutrition in SIDS.

## Figures and Tables

**Figure 1 nutrients-12-03350-f001:**
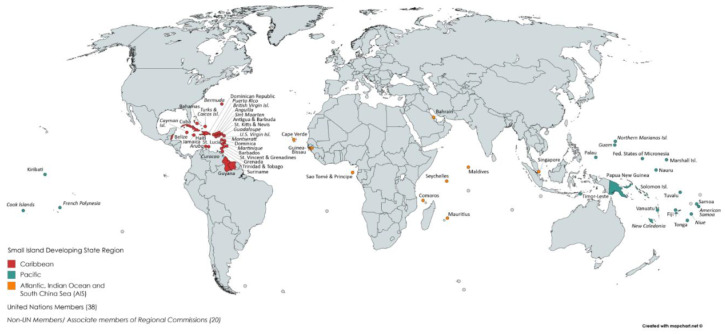
World map showing the 58 Small Island Developing States (SIDS). Countries and territories as listed at: sustainabledevelopment.un.org/topics/sids/list. (Map from Hickey and Unwin [15].)

**Figure 2 nutrients-12-03350-f002:**
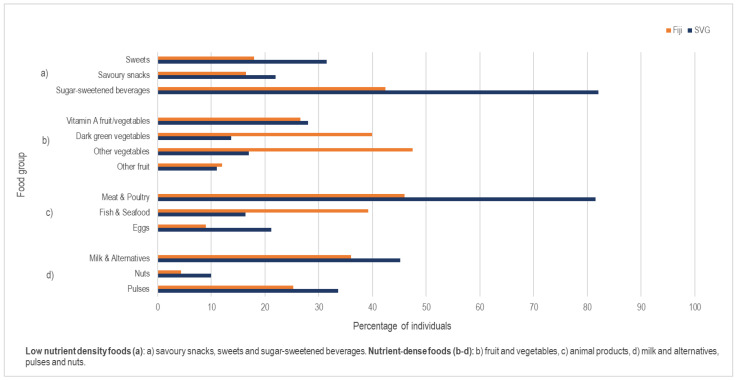
Proportion of respondents (%) in the CFaH project reporting consumption of specific food groups in one 24 h period.

**Table 1 nutrients-12-03350-t001:** Food source categories used for the Community Food and Health (CFaH) project.

Source Type	Category	Subcategory	Example
Purchase	Supermarket/Wholesaler	Wholesaler/Distributor	Larger warehouse-like spaces, offer bulk purchasing and sell larger package units, require accounts, generally sell little fresh or perishable produce, and often sell to smaller traders for re-sale.
		Supermarket	Large floor space, multiple aisles/brands, stock food and non-food/household items. Typically corporate ownership or formal franchise operations with standardised corporate image and branding. Generally centrally located (urban/semi-urban).
	Small shop	Small shop/Convenience store	Dedicated walk-in shop with a sign.
		Informal small shop	House shop/informal shop attached to or part of a home; no sign; not dedicated to food retail only; generally sell through a window.
		General dealer	Stand-alone building with a sign; trading bulk groceries and a large variety of non-food items.
		Stall/Roadside stall	Stall with shelter, display area, and sometimes storage.
		Food trucks/vans, pick-up truck or trike	Mobile vehicle including truck, van, trike.
		Mobile trader	Trolley, wheelbarrow; carry-tray, basket or buckets (other vehicles not described as van, truck or trike).
		School/College tuck shop	Dedicated shop, stall, vehicle selling any food items within school/college setting.
	Other	Informal abattoir or slaughterhouse	Abattoir/slaughterhouse
		Buy from friends or neighbours	Any purchased (not borrowed/exchanged) from friend/neighbour.
		Other	
	Food service business	Formal restaurant	Dine in, not takeaway/fast food.
		Corporate fast food shop	Run by a large corporation such as KFC, McDonalds, and Burger King.
		Informal restaurant/independent takeaway	Not run by a large corporation such as KFC, McDonalds, and Burger King; NOT a truck or stall.
		Mobile food stall/truck takeaway	Mobile vehicle of any kind that sells ready to eat food.
		Roadside bar or tavern	Predominantly selling beverages (may have a small bar menu).
Own production			Engaging in activities to produce own food such as growing, gathering, hunting, rearing or fishing.
Borrow, exchange or barter (BEB)			Includes any form of borrowing, exchanging, bartering or gifting of food or beverages.
Food aid			Includes school feeding programmes, food kitchen or parcels, food banks (such as government, non-government, community or faith-based organisations).

**Table 2 nutrients-12-03350-t002:** Characteristics of the CFaH project study populations in Fiji and SVG.

Variable	Fiji *N* = 186		SVG *N* = 147	
	*N* ^a^	*N* (%)	*N* ^a^	*N* (%)
Region	179		147	
Rural		78 (44%)		105 (71%)
Urban		101 (56%)		42 (29%)
Number of households	179	95	147	80
Household size (median (IQR))	185	4 (2−5)	146	4 (2−5)
Age (years) (mean (SD))	184	41.3 (17.0)	144	41.2 (18.3)
15 to <40 years		90 (49%)		71 (49%)
40 to <65 years		76 (41%)		54 (38%)
65 years or older		18 (10%)		19 (13%)
15 to 49 years (WRA) ^b^		81 (44%)	147	56 (38.1%)
Sex	186		143	
Male		69 (37%)		53 (37%)
Female		117 (63%)		90 (63%)
Education status	185		144	
Primary education or lower		82(44%)		77 (53%)
Secondary school completed		77 (42%)		47 (33%)
Higher education completed		26 (14%)		20 (14%)
Employment status	181		141	
Employed		54 (30%)		78 (55%)
Student		22 (12%)		9 (6%)
Homemaker		42 (23%)		3 (2%)
Retired		18 (10%)		9 (6%)
Unemployed		45 (25%)		42 (30%)
Marital status	179		143	
Never married/visiting partner		38 (21%)		70 (49%)
Married or cohabiting		119 (67%)		62 (43%)
Separated/divorced/widowed		22 (12%)		11 (8%)
Body mass index (kg/m^2^) (mean (SD))	185	29.1 (6.4)	141	28.4 (6.7)
Underweight		5 (3%)		7 (5%)
Normal weight		50 (27%)		43 (30%)
Overweight (25 to <30)		52 (28%)		39 (28%)
Obese (>30)		78 (42%)		52 (37%)
Blood pressure, hypertension	186		144	
Systolic BP (mmHg) (mean (SD))		136 (27)		121 (23)
Diastolic BP (mmHg) (mean (SD))		80 (15)		77 (13)
≥140/90		71 (38%)		29 (20%)
Hypertension ^c^		72 (39%)		41 (28%)

^a^ Number with complete data for each variable; ^b^ WRA = women of reproductive age; ^c^ blood pressure ≥ 140/90 and/or reported taking medication for a diagnosis of hypertension. SVG: Saint Vincent and the Grenadines, IQR: interquartile range, SD: standard deviation, BP: blood pressure.

**Table 3 nutrients-12-03350-t003:** Aspects of diet in the CFaH project study populations, including the dietary diversity score (DDS), minimum dietary diversity in women of reproductive age (M-DDW), and median weekly servings of selected items.

	Fiji			SVG		
	Rural	Urban	All	Rural	Urban	All
*N*	78	101	179	105	42	147
DDS (mean (SD))	3.5 (1.6)	3.7 (1.3)	3.7 (1.4)	3.6 (1.4)	4.2 (1.9)	3.8 (1.5)
Number of food groups						
<4	41%	42%	41%	52%	40%	49%
4–5	22%	45%	35%	37%	38%	37%
>5	37%	14%	24%	10%	21%	14%
M-DDW—Women aged 15 to 49 years						
N	27	35	62	42	14	56
Not met	70%	77%	74%	69%	57%	66%
Met	30%	23%	26%	31%	43%	34%
Weekly servings (median (IQR))						
Fruit	3 (2,6)	2 (1,4)	2 (1,5)	8 (3,21)	4 (1,10)	7 (2,14)
Vegetables	12 (6,14)	7 (4,10)	7 (5,14)	4 (1,7)	2 (0,6)	4 (1,7)
SSB *	2 (1,10)	3 (1,10)	3 (1,9)	8 (2,21)	7 (2,17)	7 (2,21)
Red or processed meat	2 (1,4)	2 (1,4)	2 (1,4)	2 (1,4)	4 (0,7)	2 (1,5)

* Sugar-sweetened beverages.

**Table 4 nutrients-12-03350-t004:** Social and demographic characteristics of respondents in the CFaH project by frequency of use of selected food sources in Fiji. *

Food Source	*N* (Column %)	Age (Mean (SD))	Sex (% Female)	>Primary Educ’ (%)	Household Size > 3 (%)	Region (% Rural)
Own produce						
Weekly or less	59 (31.7)	40.0 (17.8)	61.0	54.2	69.0	36.4
>Weekly	127 (68.3)	42.0 (16.7)	63.8	56.3	41.7	46.8
Diff (95% CIs)		1.9 (−3.4, 7.2)	2.8 (−12.2, 17.8)	2.1 (−13.3, 17.5)	−27.2 (−41.9, −12.6)	10.4 (−5.0, 25.9)
Supermarket/wholesaler						
Weekly or less	76 (40.9)	42.6 (18.1)	67.1	50.0	46.1	27.0
>Weekly	110 (59.1)	40.5 (16.4)	60.0	59.6	53.2	55.2
Diff (95% CIs)		−2.0 (−7.1, 3.0)	−7.1 (−21.1, 6.9)	9.6 (−4.9, 24.2)	7.2 (−7.4, 21.8)	28.2 (14.3, 42.1)
Small shop ^a^						
Weekly or less	145 (78.0)	41.2 (17.2)	64.1	55.6	50.0	43.2
>Weekly	41 (22.0)	41.8 (16.8)	58.5	56.1	51.2	45.0
Diff (95% CIs)		0.6 (−5.4, 6.6)	−5.6 (−22.6, 11.4)	0.5 (−16.7, 17.8)	1.2 (−16.1, 18.6)	1.8 (−15.6, 19.3)
Formal small shop						
Weekly or less	163 (87.6)	41.8 (17.1)	64.4	56.8	50.6	43.9
>Weekly	23 (12.4)	38.5 (17.1)	52.2	47.8	47.8	40.9
Diff (95% CIs)		−3.3 (−10.8, 4.2)	−12.2 (−33.9, 9.5)	−9.0 (−30.8, 12.8)	−2.8 (−24.6, 19.0)	−3.0 (−25.0, 18.9)
Informal small shop						
Weekly or less	150 (80.7)	41.3 (17.1)	62.7	55.7	49.7	42.4
>Weekly	36 (19.4)	41.7 (17.2)	63.9	55.6	52.8	48.6
Diff (95% CIs)		0.5 (−5.8, 6.7)	1.2 (−16.3, 18.7)	−0.1 (−18.2, 17.9)	3.1 (−15.1, 21.3)	6.2 (−12.2, 24.6)
Food service business						
Weekly or less	172 (92.5)	41.8 (17.1)	62.8	55.0	51.5	43.0
>Weekly	14 (7.5)	35.4 (16.2)	64.3	64.3	35.7	50.0
Diff (95% CIs)		−6.4 (−15.7, 2.9)	1.5 (−24.6, 27.6)	9.3 (−16.9, 35.5)	−15.7 (−41.9, 10.4)	7.0 (−20.3, 34.2)
BEB ^b^						
Weekly or less	176 (94.6)	40.7 (16.8)	61.9	56.6	52.6	42.6
>Weekly	10 (5.4)	52.0 (18.4)	80.0	40.0	10.0	60.0
Diff (95% CIs)		11.3 (0.4, 22.1)	18.1 (−7.7, 43.9)	−16.6 (−47.8, 14.7)	−42.6 (−62.6, −22.6)	17.4 (−13.9, 48.7)

* Shaded cells indicate where the 95% CIs on the difference do not cross zero; ^a^ formal and informal; ^b^ borrow, exchange or barter.

**Table 5 nutrients-12-03350-t005:** Social and demographic characteristics of respondents in the CFaH project by frequency of use of selected food sources in SVG. *

Food Source	*N* (Column %)	Age (Mean (SD))	Sex (% Female)	>Primary Educ’ (%)	Household Size > 3 (%)	Region (% Rural)
Own produce						
Weekly or less	81 (55.1)	39.4 (18.6)	65.8	50.0	51.3	58.0
>Weekly	66 (44.9)	43.3 (18.0)	59.4	42.2	51.5	87.9
Diff (95% CIs)		3.9 (−2.2, 9.9)	−6.4 (−22.4, 9.5)	−7.8 (−24.1, 8.5)	0.3 (−16.0, 16.6)	29.9 (16.5, 43.2)
Supermarket/wholesaler						
Weekly or less	42 (28.6)	44.3 (20.2)	53.7	40.0	42.9	59.5
>Weekly	105 (71.4)	40.0 (17.5)	66.7	49.0	54.8	76.2
Diff (95% CIs)		−4.3 (−11.0, 2.4)	13.0 (−4.8, 30.8)	9.0 (−8.9, 27.0)	12.0 (−5.8, 29.7)	16.7 (−0.3, 33.6)
Small shop ^a^						
Weekly or less	69 (46.9)	44.2 (20.0)	59.4	45.5	44.1	66.7
>Weekly	78 (53.1)	38.7 (16.6)	66.2	47.4	57.7	75.6
Diff (95% CIs)		−5.5 (−11.5, 0.6)	6.8 (−9.0, 22.6)	2.0 (−14.4, 18.3)	13.6 (−2.5, 29.7)	9.0 (−5.7, 23.6)
Formal small shop						
Weekly or less	92 (62.6)	43.6 (19.5)	64.1	42.7	49.5	65.2
>Weekly	55 (37.4)	37.3 (15.7)	60.8	52.7	54.5	81.8
Diff (95% CIs)		−6.3 (−12.4, −0.1)	−3.3 (−19.9, 13.3)	10.0 (−6.7, 26.8)	5.1 (−11.6, 21.8)	16.6 (2.5, 30.7)
Informal small shop						
Weekly or less	101 (68.7)	41.9 (19.1)	57.6	48.0	44.0	68.3
>Weekly	46 (31.3)	39.8 (16.7)	75.0	43.5	67.4	78.3
Diff (95% CIs)		−2.1 (−8.6, 4.4)	17.4 (1.3, 33.5)	−4.5 (−21.9, 12.9)	23.4 (6.7, 40.1)	9.9 (−5.0, 24.9)
Food service business						
Weekly or less	137 (93.2)	42.3 (18.4)	63.2	46.3	51.5	70.1
>Weekly	10 (6.8)	27.5 (10.3)	60.0	50.0	50.0	90.0
Diff (95% CIs)		−14.8 (−26.4, −3.1)	−3.2 (−34.6, 28.3)	3.7 (−28.4, 35.9)	−1.5 (−33.6, 30.6)	19.9 (−0.2, 40.0)
BEB ^b^						
Weekly or less	99 (67.4)	41.6 (18.9)	57.9	40.8	46.9	61.6
>Weekly	48 (32.7)	40.5 (17.2)	72.9	58.7	60.4	91.7
Diff (95% CIs)		−1.1 (−7.6, 5.4)	15.0 (−1.0, 31.0)	17.9 (0.6, 35.1)	13.5 (−3.5, 30.5)	30.1 (17.7, 42.4)

* Shaded cells indicate where the 95% CIs on the difference do not cross zero; ^a^ formal and informal; ^b^ borrow, exchange or barter.

**Table 6 nutrients-12-03350-t006:** Aspects of diet by frequency of use of selected food sources in the CFaH project in Fiji. *

Food Source	Mean (SD) DDS ^a^	Median No. Servings per Week			
		Fruit	Veg	SSB	Proc and Red Meat
Own produce					
Weekly or less	3.7 (1.5)	2	8	4	2
>Weekly	3.7 (1.4)	3	7	2	2
Diff (95% CIs)	0.0 (−0.5, 0.4)	1 (0, 2)	−1 (−3, 1)	−2 (−5, 1)	0 (−1, 1)
Supermarket/wholesaler					
Weekly or less	3.7 (1.4)	2	7	3	2
>Weekly	3.6 (1.4)	3	9	2	2
Diff (95% CIs)	−0.1 (−0.6, 0.3)	1 (0, 2)	2 (0, 4)	−1 (−4, 2)	0 (−1, 1)
Small shop ^b^					
Weekly or less	3.7 (1.4)	2	7	2	2
>Weekly	3.6 (1.5)	4	7	6	3
Diff (95% CIs)	−0.1 (−0.6, 0.4)	2 (1, 3)	0 (−3, 3)	4 (1, 7)	1 (0, 2)
Formal small shop					
Weekly or less	3.6 (1.4)	2	7	2	2
>Weekly	3.9 (1.6)	4	9	8	4
Diff (95% CIs)	0.3 (−0.4, 0.9)	2 (1, 3)	2 (−4, 4)	6 (2, 10)	2 (1, 3)
Informal small shop					
Weekly or less	3.7 (1.3)	2	7	2	2
>Weekly	3.5 (1.6)	4	7	6	3
Diff (95% CIs)	−0.2 (−0.7, 0.3)	2 (1, 3)	0 (−4, 4)	4 (0, 8)	1 (0, 2)
Food service business					
Weekly or less	3.7 (1.4)	2	7	2	2
>Weekly	3.5 (1.7)	4	7	8	5
Diff (95% CIs)	−0.2 (−1.0, 0.6)	2 (0, 4)	(−4, 4)	6 (1, 11)	3 (2, 4)
BEB ^c^					
Weekly or less	3.6 (1.4)	2	7	3	2
>Weekly	4.7 (1.5)	4	14	2	1
Diff (95% CIs)	1.1 (0.0, 2.2)	2 (0, 4)	7 (2, 12)	−1 (−7, 5)	−1 (−2, 0)

* Shaded cells indicate where the 95% CIs on the difference do not cross zero; ^a^ dietary diversity score; ^b^ formal and informal; ^c^ borrow, exchange or barter.

**Table 7 nutrients-12-03350-t007:** Aspects of diet by frequency of use of selected food sources in the CFaH project in SVG. *.

Food Source	Mean (SD) DDS ^a^	Median No. Servings Per Week			
		Fruit	Veg	SSB	Proc and Red Meat
Own produce					
Weekly or less	3.7 (1.4)	4	3	7	2
>Weekly	3.9 (1.7)	9	4	14	2
Diff (95% CIs)	0.3 (−0.2, 0.8)	5 (1, 9)	1 (−1, 3)	7 (1, 13)	−1 (−3, 1)
Supermarket/wholesaler					
Weekly or less	3.3 (1.5)	3	2	4	1
>Weekly	4.0 (1.5)	7	4	10	3
Diff (95% CIs)	0.6 (0.1, 1.2)	4 (0, 8)	2 (0, 4)	6 (2, 10)	2 (1, 3)
Small shop ^b^					
Weekly or less	4.1 (1.8)	7	3	7	2
>Weekly	3.5 (1.3)	7	4	14	3
Diff (95% CIs)	−0.5 (−1.0, 0.0)	0 (−4, 4)	1 (−1, 3)	7 (1, 13)	1 (−1, 3)
Formal small shop					
Weekly or less	4.1 (1.6)	7	3	7	2
>Weekly	3.3 (1.2)	6	4	14	3
Diff (95% CIs)	−0.8 (−1.3, −0.3)	−1 (−6, 4)	1 (−1, 3)	7 (1, 13)	1 (−1, 3)
Informal small shop					
Weekly or less	3.7 (1.7)	6	3	7	2
>Weekly	3.9 (1.2)	14	4	14	3
Diff (95% CIs)	0.2 (−0.4, 0.7)	8 (3, 13)	1 (−1, 3)	7 (1, 13)	1 (0, 2)
Food service business					
Weekly or less	3.8 (1.6)	7	3	7	2
>Weekly	3.8 (1.3)	9	7	21	4
Diff (95% CIs)	0.0 (−1.0, 1.0)	2 (−5, 11)	4 (0, 8)	14 (5, 23)	2 (−1, 5)
BEB ^c^					
Weekly or less	3.6 (1.6)	5	3	6	2
>Weekly	4.1 (1.3)	14	5	14	3
Diff (95% CIs)	0.5 (0.0, 1.0)	9 (5, 13)	2 (0, 4)	8 (2, 14)	1 (0, 2)

* Shaded cells indicate where the 95% CIs on the difference do not cross zero; ^a^ dietary diversity score; ^b^ formal and informal; ^c^ borrow, exchange or barter.

**Table 8 nutrients-12-03350-t008:** Factors associated with the dietary diversity score (DDS) in the CFaH project. Results of multiple linear regression with all independent variables entered together and adjustment for household sampling.

Variable	Coefficient (95% CIs)	*p* Value
Food sources used > weekly		
Supermarket/wholesaler	0.20 (−0.24, 0.65)	0.370
Small shop	−0.52 (−0.91, −0.12)	0.011
Borrow/exchange/barter	0.73 (0.21, 1.25)	0.006
Social and demographic factors		
Age (years)	0.01 (0.00, 0.03)	0.063
Female vs. male	0.22 (−0.13, 0.56)	0.214
>Primary education	0.44 (0.06, 0.82)	0.023
Household size >3	0.18 (−0.19, 0.55)	0.348
Country and region		
Rural vs. urban	−0.46 (−0.92, 0.00)	0.049
Country (SVG vs. Fiji)	0.25 (−0.21, 0.71)	0.277

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
