# Peer review of "Food Sources and Dietary Quality in Small Island Developing States: Development of Methods and Policy Relevant Novel Survey Data from the Pacific and Caribbean"

_nutrients, 2020, doi:10.3390/nu12113350_

Round 1
Reviewer 1 Report
Overall comments: There isn’t a distinct link between the first sentence and second sentence in the abstract. I am left not really understanding how the investigators are making that link. Is this toolkit new? Do they feel it should be used by others? In the abstract the sentence on line 24 – We developed a toolkit…. Seems to be out of place. The abstract is very choppy and difficult to follow. The term Small Island Developing State is capitalized in the first sentence and not capitalized in line 37. Please be consistent.
Line 43 of the introduction – lack of evidence of what? I would have liked more information in the introduction of why the two countries were chosen. Where do they fall in the continuum of the SIDS.
Why did they choose upper middle income states?
The methods lines 7-9 about obesity and diabetes in men/women in each country does not read well – reword. What is the vast majority for type 2 diabetes mean? Why not just report type 2 diabetes? Be specific in line 10 where they rank in the world. Why greater than age 15 (Lorraine is this considered the norm in international research) Was the toolkit validated with the population first and not just look at by investigators? Who completed the 24 hour recalls – these are not true 24 hour recalls since portions were not obtained other than if greater than 15 grams. This is misleading and would eliminate that wording but to say dietary diversity was measured based on the previous 24 hour intake.
Add in the equipment used to take anthropometrics. The methods talked about type 2 diabetes but no specific information on the questionnaires about this and only information on hypertension. Why discuss the prevalence in methods if no data collected? I would have expected something discussed in the results section but what is highlighted is hypertension instead.
Inconsistencies in the results with capitalization of the word table.
Discussion – line 48 not sure the connection between Brazil dietary guidelines and the countries being studied.
Conclusion – the study aim was to talk about nutrition related diseases and this wasn’t brought into the conclusion.
Reviewer 2 Report
Dear authors,
Thank you for the opportunity to review this manuscript describing the development and use of a toolkit to assess relationships between diet, sources of food consumed and risk of nutrition related NCD, and individual associations between socio-demographic characteristics, food sources, and aspects of diet in Fiji and SVG.
The manuscript is novel, and contributes to the limited work in these settings. Please find my comments below:
Abstract
- Suggest changing nutrition-related disease to diet-related disease. A reference to diet is made in the aim, so it would be more appropriate to use this. Diet screener is also used in the methods (also see line 15 in intro and in discussion).
- Line 26 – eligible for? To participate?
Introduction
- Line 4 – it would be useful to see the proportion of these independent SIDS (i.e. how many of the 29 are in the Pacific?).
- The introduction, while stating key points, could be edited to improve flow. Editing could help minimise repetition, particularly when referring to the Caribbean and Pacific (i.e. line 17).
Methods
- Suggest including maps to show where these countries are located, this would be particularly useful for readers unfamiliar with these areas.
- While urban/rural classifications are used, could the areas where data collection occurred be shown on a map? This would assist with further research efforts to ensure duplication does not occur and aid comparison of data.
- When considering dietary diversity, were the same food groups used for each country, and if so, how were country specific differences considered?
- Line 26 – missing a bracket end
- Diet screener - please provide more detail on how these questions were presented.
- Line 36 – were these follow up questions standardised?
- Was the data collected in local language or English, how was translation accounted for?
- Ethics; line 21 should be ‘provided’
Results
- Overall, clear and easy to navigate.
- Could you explain why the age categories were chosen as they are?
- Table 3. Not clear what no/yes for M-DDW is – assuming this is proportion who met?
Discussion
The discussion situates the work, however it is disappointing to see that there is a lack of recent Caribbean/Pacific specific findings being referred to: i.e. for the Pacific:
- O’Meara L, Williams SL, Hickes D, Brown P. Predictors of Dietary Diversity of Indigenous Food-Producing Households in Rural Fiji. Nutrients. 2019 Jul;11(7):1629.
- Horsey B, Swanepoel L, Underhill S, Aliakbari J, Burkhart S. Dietary diversity of an adult Solomon Islands population. Nutrients. 2019 Jul;11(7):1622.
- Eme PE, Burlingame B, Douwes J, Kim N, Foliaki S. Quantitative estimates of dietary intake in households of South Tarawa, Kiribati. Asia Pacific journal of clinical nutrition. 2019 Mar;28(1):131.
- Hidalgo DM, Witten I, Nunn PD, Burkhart S, Bogard JR, Beazley H, Herrero M. Sustaining healthy diets in times of change: linking climate hazards, food systems and nutrition security in rural communities of the Fiji Islands. Regional Environmental Change. 2020;20:73.
Line 40. Suggest amending this line - Additionally, in SVG, but not in Fiji, regular BEB of food was strongly associated with consumption of own produced food. This suggests, especially in SVG, that a high proportion of BEB food was ‘own produced’. Remove, ‘especially’ as this contradicts the previous sentence.
The reference to Covid-19 seems to be out of place in the discussion (and not relevant to the age of the data (2018)).
‘Obesogenic food’ is not recognised term, suggest either leaving as ultra-processed, and or discussing the environment in respect to being obesogenic.
Overall, this manuscript makes a valuable contribution to the literature, but could be improved with minor editing, and incorporating region specific links in the discussion.
Thank you,
